# The impact of *cytochrome P450 3A* genetic polymorphisms on tacrolimus pharmacokinetics in ulcerative colitis patients

**Maizumi Furuse**[1], **Shuhei Hosomi**[1]*, **Yu Nishida**[1], **Shigehiro Itani**[1], **Yuji Nadatani**[1], **Shusei Fukunaga**[1], **Koji Otani**[1], **Fumio Tanaka**[1], **Yasuaki Nagami**[1], **Koichi Taira**[1], **Noriko Kamata**[1], **Toshio Watanabe**[1], **Kenji Watanabe**[2], **Yasuhiro Fujiwara**[1]

1 Department of Gastroenterology, Osaka City University Graduate School of Medicine, Osaka, Japan,
2 Department of Center for Inflammatory Bowel Disease, Division of Internal Medicine, Hyogo College of Medicine, Hyogo, Japan

* m1265271@med.osaka-cu.ac.jp

**Data Availability Statement:** All relevant data are within the paper and its Supporting information files.

## Abstract

Tacrolimus (Tac) is an effective remission inducer of refractory ulcerative colitis (UC). Gene polymorphisms result in interindividual variability in Tac pharmacokinetics. In this study, we aimed to examine the relationships between gene polymorphisms and the metabolism, pharmacokinetics, and therapeutic effects of Tac in patients with UC. Forty-five patients with moderate-to-severe refractory UC treated with Tac were retrospectively enrolled. Genotyping for cytochrome P450 (*CYP) 3A4\*1G*, *CYP3A5\*3*, *CYP2C19\*2*, *CYP2C19\*3*, nuclear receptor subfamily 1 group I member 2 (*NR1I2*)–*25385C>T*, *ATP-binding cassette subfamily C member 2* (*ABCC2*)–*24C>T*, *ABCC2 1249G>A*, and *ABCC2 3972C>T* was performed. Concentration/dose (C/D) ratio, clinical therapeutic effects, and adverse events were evaluated. The C/D ratio of Tac in UC patients with the *CYP3A4\*1G* allele was statistically lower than in those with the *CYP3A4\*1/\*1* allele (P = 0.005) and significantly lower in patients with *CYP3A5\*3/\*3* than in those with *CYP3A5\*1* (P < 0.001). Among patients with the *CYP3A4\*1G* allele, the C/D ratio was significantly lower in patients with *CYP3A5\*1* than in those with *CYP3A5\*3/\*3* (P = 0.001). Patients with the *NR1I2–25385C/C* genotype presented significantly more overall adverse events than those with the *C/T* or *T/T* genotype (P = 0.03). Although *CYP3A4\*1G* and *CYP3A5\*3* polymorphisms were related to Tac pharmacokinetics, *CYP3A5* presented a stronger effect than *CYP3A4*. The *NR1I2–25385C/C* genotype was related to the overall adverse events. The evaluation of these polymorphisms could be useful in the treatment of UC with Tac.

## Introduction

Ulcerative colitis (UC), a chronic inflammatory disease of unknown etiology, occurs in the colorectum [1]. Although corticosteroid therapy is the mainstay option for inducing UC remission, 33% of severe active UC cases have been reported to be refractory to corticosteroids [2]. Calcineurin inhibitors such as cyclosporine and tacrolimus (Tac) [3] are effective remission

**Funding:** The authors received no specific funding for this work.

**Competing interests:** The authors have declared that no competing interests exist.

inducers for refractory UC. The effective and safe doses of Tac differ among individuals as the therapeutic range of Tac is narrow. Therefore, close monitoring of the plasma drug concentration is necessary.

Cytochrome P-450 (CYP) 3A, a major member of the CYP enzyme subfamily, is predominantly responsible for the metabolism of Tac via the demethylation of 13-o-demethyltacrolimus [4]. The mRNAs and proteins of CYP3A4 and CYP3A5, two major enzymes of the CYP3A subfamily, are abundantly expressed in the adult liver [5]. Interindividual variability in Tac metabolism is mainly due to single nucleotide polymorphisms (SNPs) in CYP3A5, and dosing recommendations for Tac based on the CYP3A5 genotype have been published in the field of organ and stem cell transplantation [6]. A SNP at position 6986A within intron 3 of *CYP3A5*, which is referred to as *CYP3A5\*3* (rs776746, 6986A > G), can reduce the expression of the functional CYP3A5 protein because of a splicing disorder [7]. Therefore, individuals with at least one *CYP3A5\*1* allele are known as CYP3A5 expressers and those with *CYP3A5\*3/\*3* genotype are known as CYP3A5 non-expressers [8]. CYP3A5 expressers have lower dose-adjusted Tac blood concentrations and require higher Tac doses than CYP3A5 non-expressers [9–11]. Despite these findings, the interindividual variability in Tac pharmacokinetics has not been adequately explained [12, 13], as the pharmacokinetics vary among CYP3A5 expressers and non-expressers. Therefore, polymorphisms in other candidate genes have been investigated to explain the interindividual variability in Tac pharmacokinetics [14].

*CYP3A4*, which encodes CYP3A4, another major enzyme of the CYP3A subfamily involved in drug pharmacokinetics, has several SNPs related to the metabolic activity of CYP3A4 [15, 16], such as the *CYP3A4\*1B* (rs2740574, −392A>G) [17] and *CYP3A4\*1G* (rs2242480, 20230C>T) alleles with increased CYP3A4 enzyme activity [18–20] and the *CYP3A4\*22* (rs35599367, 15389C>T) allele with reduced CYP3A4 enzyme activity [21]. The mutation frequency of *CYP3A4\*1G* is relatively high in the Japanese population [22], but Japanese individuals do not have the *CYP3A4\*1B* and *CYP3A4\*22* alleles [9, 21]. Other gene polymorphisms related to drug metabolism can influence the pharmacokinetics of Tac. Pregnane X receptor (PXR, encoded by *NR1I2*) is involved in the transcriptional regulation of CYP enzymes, including CYP3A4 and CYP3A5. In fact, the induction of CYP3A4 by the *NR1I2* −25385T (rs3814055, −25385C>T) allele has been reported to be higher than that by the *NR1I2* −25385C/C allele [23], resulting in lower Tac concentrations [24]. Although CYP2C19 may not directly affect Tac pharmacokinetics, *CYP2C19* polymorphisms can affect the pharmacokinetics of Tac when co-administered with voriconazole [25, 26]. Regarding drug efflux, a recent study showed that the polymorphism of *ABCC2*, which encodes multidrug resistance-associated protein 2 that plays a role in Tac efflux into the lumen in association with CYP3A in the small intestine [27], can affect Tac pharmacokinetics [28]. However, the relationship between these SNPs, except *CYP3A5\*3*, and the pharmacokinetics and effects of Tac in patients with UC has not been clarified. Therefore, the aim of this study was to investigate the relationship of *CYP3A4*, *CYP2C19*, *NR1I2*, and *ABCC2* polymorphisms as well as *CYP3A5\*3/\*3* with the pharmacokinetics and therapeutic effects of Tac in patients with UC. Furthermore, we examined the relationship between gene polymorphisms and adverse events in patients with UC.

## Materials and methods

### Patient selection

We retrospectively enrolled 47 Japanese patients with moderate-to-severe UC. All patients were treated with Tac for remission induction in the University Hospital between January 2009 and January 2018. Demographic characteristics, laboratory results, and medication history were obtained from the medical records.

## Treatment protocol and evaluation of treatment efficacy

The initial Tac dose was generally 0.05 mg/kg/day and orally administered twice a day at 12-h intervals. Blood Tac concentration was measured by either an affinity column-mediated immunoassay (from January 2009 to January 2013), a chemiluminescent immunoassay (from February 2013 to June 2014), or an electro-chemiluminescence immunoassay (July 2014 to January 2018) in the in-hospital laboratory. In most patients, to monitor the blood Tac concentration, trough Tac level was measured three times a week in the first 2 weeks. The dose was adjusted to achieve a high trough level of 10–15 ng/mL in the first 2 weeks. Two weeks after administration, the dose was adjusted to achieve a low trough level of 5–10 ng/mL.

Clinical disease activity was assessed using the partial Mayo score (sum of 3 subscores of the Mayo score without the endoscopic findings) [29] at 1, 2, and 4 weeks. Clinical response was defined as a reduction in the partial Mayo score by ≥2 points accompanied by a decrease of at least 30% from baseline and a decrease in the rectal bleeding subscore of ≥1 or an absolute rectal bleeding subscore of 0 or 1. Safety was evaluated using physical findings and blood tests based on medical records for 12 weeks.

## Ethical considerations

This study was approved by the Ethics Committee of Osaka City University Graduate School of Medicine (approval number: 3293). Written informed consent was obtained from all patients at the start of this study. All data were fully anonymized before we accessed them.

## Genotyping

DNA was isolated from the patients' peripheral blood samples in ethylenediaminetetraacetic acid using the QIAamp® DNA Blood Mini Kit (Qiagen, Hilden, Germany). We selected the SNPs that have been reported to potentially affect the pharmacokinetics of Tac, as described in the Introduction. Finally, genotyping was performed for the following nine SNPs: *CYP3A4* rs2242480 C>T (*CYP3A4*1G* allele), *CYP3A4* rs4646438 –>T (*CYP3A4*6* allele), *CYP3A5* rs776746 C>T (*CYP3A5*3* allele), *CYP2C19* rs4244285 G>A (*CYP2C19*2* allele), *CYP2C19* rs4986893 G>A (*CYP2C19*3* allele), *NR1I2* rs3814055 C>T, *ABCC2* rs717620 C>T (–24), *ABCC2* rs2273697 G>A (1249), and *ABCC2* rs3740066 C>T (3972). It was performed using TaqMan® SNP genotyping assay kit (Applied Biosystems, Foster City, CA, USA) according to the manufacturer's instructions.

## Statistical analysis

The concentration and dose ratio (C/D ratio) [(ng/mL)/(mg/kg)] were calculated as Tac trough levels (ng/mL) multiplied by body weight (kg) and divided by Tac dose (mg), and it was used as an index of Tac metabolism.

All SNPs were tested for deviation from the Hardy–Weinberg equilibrium and $p > 0.05$ (chi-squared test) was considered to indicate equilibrium. For pairwise linkage disequilibrium (LD) analysis, $r^2$ was calculated using Haploview software (Broad Institute, Cambridge, MA).

Continuous variables are summarized as mean and standard deviation (SD). Unpaired *t*-test was used to assess the differences in mean values. Chi-squared test or Fisher's exact test was performed to evaluate the differences in clinical data between each group. Fisher's exact test was applied to small samples. Multivariate analyses were performed using a linear logistic regression model to identify factors associated with the therapeutic effect of Tac. The association between gene polymorphisms and Tac pharmacokinetics was analyzed without and with correction for age and the partial Mayo score. These statistical analyses were performed using

EZR (Saitama Medical Center, Jichi Medical University), a graphical user interface for R (The R Foundation for Statistical Computing, version 2.13.0). Results with a p value of <0.05 were considered statistically significant.

## Results

### Baseline characteristics

There were 47 patients with moderate-to-severe UC; they were all Japanese. Of these patients, two were excluded owing to follow-up loss. Finally, 45 patients were eligible for this retrospective study.

Table 1 shows the baseline characteristics of the 45 patients, including 24 men (53%). The mean age was 43 years and the mean disease duration was 6 years. The number of patients who received systemic corticosteroids before starting Tac was 30 (67%). The mean corticosteroid dose was 22 mg. Other treatments were antitumor necrosis factor (TNF)-α antibodies in seven patients (16%) and thiopurines in eight patients (18%). The mean partial Mayo score was 7.

### Frequency of genotypes

Table 2 shows the allele and genotype frequencies of the nine SNPs. Finally, we analyzed eight SNPs in this study because no patient had the *CYP3A4*6* allele. The allele frequencies of the eight SNPs did not deviate from the Hardy–Weinberg equilibrium. However, there was a strong degree of pairwise LD between the *CYP3A5*3* and *CYP3A4*1G* polymorphisms ($r^2 = 0.859$).

**Table 1. Baseline characteristics of patients.**

|  | All patients |
|---|---|
| Number of patients | 45 |
| Age, mean ± SD | 43 ± 16.9 |
| Male, n (%) | 24 (53) |
| Body weight, mean ± SD, kg | 55 ± 9.7 |
| Disease duration, mean ± SD, years | 6 ± 7.9 |
| Disease location |  |
| Pancolitis, n (%) | 34 (76) |
| Left-sided colitis, proctitis, n (%) | 11 (24) |
| Response to corticosteroid therapy |  |
| Resistant, n (%) | 19 (42) |
| Dependent, n (%) | 26 (58) |
| Corticosteroid dose, mean ± SD, mg[†] | 22 ± 23.1 |
| Anti-TNF-α antibody (or biologics) refractory disease, n (%) | 7 (16) |
| Thiopurine treatment, n (%) | 8 (18) |
| Food intake (yes/no) | 6/39 |
| Hemoglobin, mean ± SD, g/dL | 12 ± 2.1 |
| Serum albumin, mean ± SD, g/dL | 3 ± 0.7 |
| C-reactive protein, mean ± SD, mg/dL | 4 ± 5.1 |
| Partial Mayo score before starting Tac therapy, mean ± SD | 7 ± 1.3 |

[†]The listed value is the prednisolone equivalent.

SD: standard deviation; TNF: tumor necrosis factor

**Table 2. Allele and genotype frequencies.**

| Gene | dbSNP | Position | Allele | Allele frequency (%) | Genotype | n | | Frequency (%) |
|------|-------|----------|--------|----------------------|----------|---|---|---------------|
| *CYP3A4* | 2242480 | 20230 | G (*1) | 81 | *1/*1 | 30 | *1/*1 | 67 |
| | | | A (*1G) | 19 | *1/*1G | 13 | *1/*1G + *1G/*1G | 33 |
| | | | | | *1G/*1G | 2 | | |
| *CYP3A4* | 4646438 | 99766411 | – | 45 | –/– | 45 | | |
| | | | A (*6) | 0 | –/*6 | 0 | | |
| | | | | | *6/*6 | 0 | | |
| *CYP3A5* | 776746 | 6986 | A (*1) | 17 | *1/*1 | 2 | Expresser (*1/*1 + *1/*3) | 29 |
| | | | G (*3) | 83 | *1/*3 | 11 | Non-expresser (*3/*3) | 71 |
| | | | | | *3/*3 | 32 | | |
| *CYP2C19* | 4244285 | 681 | G | 80 | GG | 28 | GG | 62 |
| | | | A | 20 | GA | 16 | GA + AA | 38 |
| | | | | | AA | 1 | | |
| *CYP2C19* | 4986893 | *3 | G | 89 | GG | 35 | GG | 78 |
| | | | A | 11 | GA | 10 | GA + AA | 22 |
| | | | | | AA | 0 | | |
| *ABCC2* | 717620 | –24 | C | 84 | CC | 32 | CC | 71 |
| | | | T | 16 | CT | 12 | CT + TT | 29 |
| | | | | | TT | 1 | | |
| *ABCC2* | 2273697 | 1249 | G | 84 | GG | 34 | GG | 76 |
| | | | A | 16 | GA | 8 | GA + AA | 24 |
| | | | | | AA | 3 | | |
| *ABCC2* | 3740066 | 3972 | C | 81 | CC | 30 | CC | 67 |
| | | | T | 19 | CT | 13 | CT + TT | 33 |
| | | | | | TT | 2 | | |
| *NR1I2* | 3814055 | –25385 | C | 76 | CC | 26 | CC | 58 |
| | | | T | 24 | CT | 16 | CT + TT | 42 |
| | | | | | TT | 3 | | |

ABCC2, ATP-binding cassette subfamily C member 2; CYP2C19: cytochrome P450 family 2 subfamily C member 19; CYP3A4: cytochrome P450 family 2 subfamily A member 4; CYP3A5: cytochrome P450 family 2 subfamily A member 5; dbSNP: the single nucleotide polymorphism database; NR1I2, nuclear receptor subfamily 1 group I member 2

As previously reported [8], the *CYP3A5* genotype was divided into *CYP3A5*1/*1+*1/*3* (expressers) and *CYP3A5*3/*3* (non-expressers). The *CYP3A4* genotype was divided into *CYP3A4*1/*1* and *CYP3A*1/*1G+*1G/*1G*, as reported previously [18].

## Association of gene polymorphisms with tacrolimus pharmacokinetics

To reveal the association between these gene polymorphisms and Tac pharmacokinetics, we investigated the days required to achieve the target trough level, daily tacrolimus dose, and C/D ratio in the high trough phase (Table 3).

There were no significant associations of *NR1I2*, *CYP2C19*, and *ABCC2* polymorphisms with the C/D ratio in the high trough phase. Patients with the *CYP3A4*1G* allele required significantly longer duration and higher Tac daily dose to achieve the target trough levels than those with the *CYP3A4*1/*1* allele (P = 0.002 and P = 0.019, respectively). The C/D ratio was significantly lower in patients with the *CYP3A4*1G* allele than in those with the *CYP3A4*1/*1* allele (P = 0.005). Regarding *CYP3A5* polymorphism, CYP3A5 expressers required

**Table 3. Relationship between genetic polymorphism and pharmacokinetics.**

| Gene | Position | Genotype | n | Days required to achieve target trough levels[†] | | Daily dose when reached high trough[‡] | | C/D ratio when reached high trough | |
|---|---|---|---|---|---|---|---|---|---|
| | | | | day, mean ± SD | P value | mg/kg, mean ± SD | P value | ng/mL per mg/kg, mean ± SD | P value |
| *CYP3A4* | *1G | *1/*1 | 30 | 6.9 ± 2.6 | 0.002[##] | 4.3 ± 3.1 | 0.019[#] | 146.5 ± 68.1 | 0.005[##] |
| | | *1/*1G + *1G/*1G | 15 | 10.9 ± 4.0 | | 7.0 ± 3.8 | | 87.9 ± 46.1 | |
| *CYP3A5* | *3 | *1/*1 + *1/*3 | 13 | 7.8 ± 3.6 | 0.002[##] | 11.7 ± 3.8 | < 0.0001[###] | 74.9 ± 32.6 | < 0.001[###] |
| | | *3/*3 | 32 | 4.2 ± 3.1 | | 6.8 ± 2.6 | | 148.2 ± 66.3 | |
| *CYP2C19* | *2 | GG | 28 | 5.8 ± 3.9 | 0.17 | 7.7 ± 2.8 | 0.21 | 134.9 ± 67.9 | 0.32 |
| | | GA + AA | 17 | 4.3 ± 3.0 | | 9.1 ± 4.7 | | 114.0 ± 66.0 | |
| *CYP2C19* | *3 | GG | 35 | 5.4 ± 4.0 | 0.66 | 8.4 ± 3.9 | 0.54 | 131.0 ± 73.4 | 0.46 |
| | | GA + AA | 10 | 4.8 ± 1.7 | | 7.6 ± 2.8 | | 112.9 ± 38.2 | |
| *ABCC2* | −24 | CC | 32 | 4.9 ± 3.1 | 0.33 | 7.9 ± 3.9 | 0.39 | 129.0 ± 71.7 | 0.76 |
| | | CT + TT | 13 | 6.1 ± 4.6 | | 9.0 ± 3.0 | | 122.0 ± 56.8 | |
| *ABCC2* | 1249 | GG | 34 | 5.5 ± 3.8 | 0.36 | 8.3 ± 3.8 | 0.80 | 130.5 ± 64.7 | 0.54 |
| | | GA + AA | 11 | 4.4 ± 2.9 | | 8.0 ± 3.3 | | 116.2 ± 76.6 | |
| *ABCC2* | 3972 | CC | 30 | 4.9 ± 3.2 | 0.42 | 7.9 ± 4.0 | 0.33 | 129.5 ± 72.1 | 0.73 |
| | | CT + TT | 15 | 5.9 ± 4.3 | | 9.0 ± 2.8 | | 122.1 ± 58.2 | |
| *NR1I2* | −25385 | CC | 26 | 5.5 ± 3.2 | 0.59 | 9.1 ± 3.9 | 0.075 | 117.8 ± 66.4 | 0.29 |
| | | CT + TT | 19 | 4.9 ± 4.2 | | 7.1 ± 3.0 | | 139.6 ± 68.0 | |

[#] = P < 0.05,

[##] = P < 0.01,

[###] = P < 0.001

[†]Target trough levels mean high trough level of 10 to 15 ng/mL for first two weeks low trough level of 5 to 10 ng/mL after 2 weeks of administration.

[‡]High trough means trough level more than 10 ng/ml.

ABCC2: ATP binding cassette subfamily C member 2; C/D ratio: Tacrolimus concentration and dose ratio; CYP2C19: cytochrome P450 family 2 subfamily C member 19; CYP3A4: cytochrome P450 family 2 subfamily A member 4; CYP3A5: cytochrome P450 family 2 subfamily A member 5; NR1I2, nuclear receptor subfamily 1 group I member 2

significantly longer duration and higher Tac daily dose to achieve the target trough level than CYP3A5 non-expressers (P = 0.002 and P < 0.0001, respectively). The C/D ratio was significantly lower in CYP3A5 expressers than in CYP3A5 non-expressers (P < 0.001).

Furthermore, we investigated the interaction effect between *CYP3A4*1G* and *CYP3A5*3* polymorphisms on Tac pharmacokinetics, as there was a strong degree of pair-wise LD between *CYP3A5*3* and *CYP3A4*1G* polymorphisms (S1 Table). Among CYP3A5 expressers, none had *CYP3A4*1/*1*. Among 15 patients with the *CYP3A4*1G* allele, the C/D ratio was significantly lower in CYP3A5 expressers than in CYP3A5 non-expressers (P = 0.001). This suggests that *CYP3A5* polymorphism could have a stronger effect on Tac pharmacokinetics than *CYP3A4* polymorphism.

### Relationship between *CYP3A4*1G* and *CYP3A5*3* polymorphisms and therapeutic effects

We evaluated the therapeutic effects of Tac at 1, 2, and 4 weeks in patients with each genetic polymorphism. There were no significant associations between *NR1I2*, *CYP2C19*, and *ABCC2* polymorphisms and therapeutic effects (data not shown). Patients with *CYP3A4*1/*1* showed a significantly higher therapeutic effect than those with *CYP3A4*1/*1G+*1G/*1G* at 1 week after Tac initiation (Fig 1A). Similarly, CYP3A5 non-expressers showed a significantly higher therapeutic effect than CYP3A5 expressers at 1 week after Tac initiation (Fig 1B).

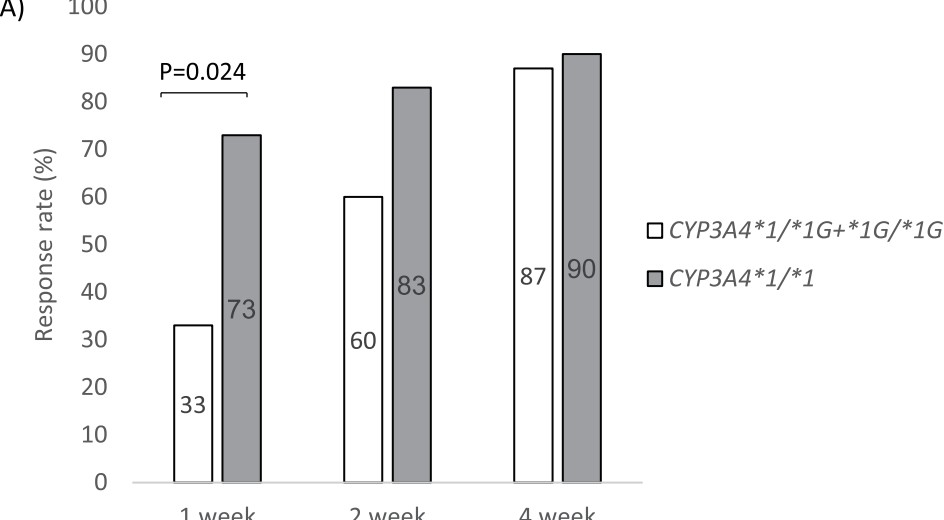

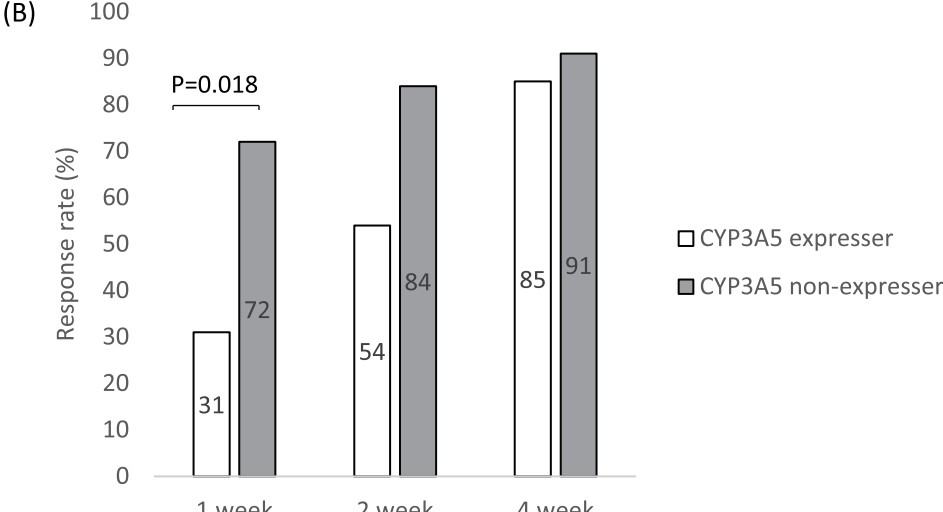

**Fig 1. Response rate of patients with *CYP3A4* and *CYP3A5* polymorphisms after tacrolimus initiation.** Panel A shows the response rate of patients with *CYP3A4* polymorphisms at 1, 2, and 4 weeks after Tac treatment, and panel B shows that of patients with *CYP3A5* polymorphisms. Clinical response was defined as a partial Mayo score reduction of ≥2 points accompanied by a decrease of at least 30% from baseline and a decrease in the rectal bleeding subscore of ≥1 or an absolute rectal bleeding subscore of 0 or 1. CYP3A4: cytochrome P450 family 2 subfamily A member 4; CYP3A5: cytochrome P450 family 2 subfamily A member 5.

*CYP3A4* and *CYP3A5* polymorphisms, patient background, concomitant thiopurine treatment, and laboratory data before Tac induction were analyzed to identify factors affecting the therapeutic effects of Tac. The univariate logistic regression analysis indicated *CYP3A4* and *CYP3A5* polymorphisms as significant factors involved in the therapeutic effect of Tac (Table 4). For these polymorphisms, a multivariate logistic regression model after adjusting for age and p-Mayo revealed both *CYP3A4* and *CYP3A5* polymorphisms as significant factors affecting the therapeutic effects of Tac with similar odds ratios (ORs): *CYP3A4*1/*1* (OR: 0.15,

**Table 4. Univariate analysis of factors associated with efficacy at 1 week after tacrolimus treatment.**

| | n | Case (%) | OR (95% CI) | P value |
|---|---|---|---|---|
| Age | 45 | | 1.01 (0.98–1.05) | 0.54 |
| Gender | | | | 0.72 |
| Male | 24 | 15 (63) | 1.00 | |
| Female | 21 | 12 (57) | 0.80 (0.24–2.64) | |
| Body weight | 45 | | 0.97 (0.91–1.03) | 0.30 |
| Disease duration | 45 | | 1.00 (1.00–1.00) | 0.98 |
| Disease location | | | | 0.33 |
| Pancolitis | 34 | 19 (56) | 1.00 | |
| Left-sided colitis, proctitis | 11 | 8 (73) | 2.11 (0.48–9.34) | |
| Response to corticosteroids | | | | 1.00 |
| Resistant | 20 | 12 (60) | 1.00 | |
| Dependent | 25 | 15 (60) | 1.00 (0.30–3.32) | |
| Corticosteroids dose | 45 | | 1.00 (0.98–1.03) | 0.97 |
| Anti-TNF-α antibody (or biologics) refractory disease | | | | 0.51 |
| Yes | 7 | 5 (71) | 1.00 | |
| No | 38 | 22 (58) | 0.55 (0.10–3.2) | |
| Thiopurine treatment | | | | 0.53 |
| Yes | 8 | 4 (50) | 1.00 | |
| No | 37 | 23 (62) | 1.64 (0.35–7.64) | |
| Hemoglobin | 45 | | 0.91 (0.68–1.23) | 0.54 |
| Serum albumin | 45 | | 0.69 (0.27–1.77) | 0.45 |
| C-reactive protein | 45 | | 0.94 (0.84–1.06) | 0.32 |
| Partial Mayo score before starting tacrolimus therapy | 45 | | 1.61 (0.92–2.81) | 0.09 |
| Genotype | | | | |
| *CYP3A4* | | | | 0.013[#] |
| *1/*1 | 30 | 22 (73) | 1.00 | |
| *1/*1G+*1G/*1G | 15 | 5 (33) | 0.18 (0.05–0.70) | |
| *CYP3A5* | | | | 0.015[#] |
| *3/*3 (non-expresser) | 32 | 23 (72) | 1.00 | |
| *1/*1+*1/*3 (expresser) | 13 | 4 (31) | 0.17 (0.04–0.71) | |

[#] = P < 0.05,

[##] = P < 0.01,

[###] = P < 0.001

CI: confidence interval; CYP3A4: cytochrome P450 family 2 subfamily A member 4; CYP3A5: cytochrome P450 family 2 subfamily A member 5; OR: odds ratio

95% confidence interval (CI): 0.04–0.64, P = 0.010) and CYP3A5 non-expressers (OR: 0.16, 95% CI: 0.04.0.71, P = 0.016) (Table 5).

## Association between gene polymorphisms and adverse events

Fifty-three adverse events were observed over 12 weeks. There were no serious adverse events. The most frequent events were renal impairment (n = 14), followed by hypomagnesemia (n = 12), hyperkalemia (n = 6), and neuropathy (n = 5) (S2 Table). Individuals with the *NR1I2–25385C/C* genotype presented significantly more events than those with the *C/T* or *T/T* genotype for all events (P = 0.03) (Table 6). There was no association between the other gene polymorphisms and adverse events.

Table 5. **Multivariate analysis of factors associated with efficacy at 1 week after tacrolimus treatment.**

| | Without correction | | | Correction for age and partial Mayo score | | |
|---|---|---|---|---|---|---|
| | OR | 95% CI | P value | OR | 95% CI | P value |
| *CYP3A4* | 0.18 | 0.05–0.70 | 0.013[#] | 0.15 | 0.04–0.64 | 0.010[#] |
| *1/*1G+*1G/*1G vs *1/*1 | | | | | | |
| *CYP3A5* | 0.17 | 0.04–0.71 | 0.015[#] | 0.16 | 0.04–0.71 | 0.016[#] |
| *1/*1+*1/*3 vs *3/*3 | | | | | | |

[#] = P < 0.05,

[##] = P < 0.01,

[###] = P < 0.001

CI: confidence interval; CYP3A4: cytochrome P450 family 2 subfamily A member 4; CYP3A5: cytochrome P450 family 2 subfamily A member 5; OR: odds ratio

## Discussion

This retrospective study showed that *CYP3A4*1G* and *CYP3A5*3* polymorphisms affected the pharmacokinetics and therapeutic effect of Tac in patients with UC. The effect of *CYP3A5* polymorphism on Tac pharmacokinetics was strong compared with that of *CYP3A4* polymorphism. Furthermore, the *NR1I2–25385* genotype was related to the overall adverse events, implicating that this polymorphism might be a potential predictor of the adverse events of Tac therapy.

Table 6. **Association between gene polymorphisms and adverse events.**

| Gene | Position | Genotype | n | All adverse events | | Renal impairment | | Hypomagnesemia | |
|---|---|---|---|---|---|---|---|---|---|
| | | | | n (%) | P value | n (%) | P value | n (%) | P value |
| *CYP3A4* | *1G | *1/*1 | 30 | 21 (70) | 0.53 | 9 (30) | 1.00 | 6 (20) | 0.29 |
| | | *1/*1G + *1G/*1G | 15 | 9 (60) | | 5 (33) | | 6 (40) | |
| *CYP3A5* | *3 | *1/*1 + *1/*3 | 13 | 7 (54) | 0.50 | 5 (38) | 0.50 | 4 (31) | 0.72 |
| | | *3/*3 | 32 | 22 (69) | | 9 (28) | | 8 (25) | |
| *CYP2C19* | *2 | GG | 28 | 19 (68) | 1.00 | 9 (32) | 1.00 | 6 (21) | 0.28 |
| | | GA + AA | 17 | 11 (65) | | 5 (29) | | 5 (29) | |
| | *3 | GG | 35 | 24 (69) | 0.71 | 11 (31) | 1.00 | 11 (31) | 0.09 |
| | | GA + AA | 10 | 6 (6) | | 3 (30) | | 0 | |
| *ABCC2* | –24 | CC | 32 | 22 (69) | 0.73 | 10 (31) | 1.00 | 7 (22) | 0.70 |
| | | CT + TT | 13 | 8 (62) | | 4 (31) | | 4 (31) | |
| | 1249 | GG | 34 | 22 (65) | 0.73 | 10 (29) | 0.72 | 8 (24) | 1.00 |
| | | GA + AA | 11 | 8 (73) | | 4 (36) | | 3 (27) | |
| | 3972 | CC | 30 | 21 (70) | 0.52 | 9 (30) | 1.00 | 7 (23) | 1.00 |
| | | CT + TT | 15 | 9 (60) | | 5 (33) | | 4 (27) | |
| *NR1I2* | –25385 | CC | 26 | 21 (81) | 0.03[#] | 9 (35) | 0.75 | 6 (23) | 1.00 |
| | | CT + TT | 19 | 9 (47) | | 5 (26) | | 5 (26) | |

[#] = P < 0.05,

[##] = P < 0.01,

[###] = P < 0.001

ABCC2: ATP-binding cassette subfamily C member 2; CYP2C19: cytochrome P450 family 2 subfamily C member 19; CYP3A4: cytochrome P450 family 2 subfamily A member 4; CYP3A5: cytochrome P450 family 2 subfamily A member 5; NR1I2: nuclear receptor subfamily 1 group I member 2

Approximately 15% of UC patients have been reported to develop an acute severe colitis, which sometimes requires urgent / emergency surgery owing to major complications such as perforation, toxic megacolon, and massive hemorrhage [30]. The therapeutic range of the Tac trough level, which is 10–15 ng/mL for an induction period of 1 week in Japan [31], is narrow; therefore, close monitoring of the Tac trough level is necessary to achieve the optimum concentration as fast as possible and reduce the risk of emergency colectomy. Recently, Okabayashi et al. showed that the individualized dosage adjustment of Tac based on *CYP3A5*3* polymorphisms could be useful to quickly achieve a high Tac trough level and early therapeutic efficacy [32]. The principle of this strategy is that Tac is primarily metabolized by the biotransformation enzymes CYP3A4 and CYP3A5 in the liver and gut, and this affects its blood concentration [4]; moreover, *CYP3A5* polymorphisms were used because they are reported to affect the pharmacokinetics of Tac. Furthermore, organ transplantation studies have reported that CYP3A5 expressers (*CYP3A5*1/*1+*1/*3*) have a significantly higher Tac metabolic capacity than *CYP3A5* non-expressers *(CYP3A5*3/*3)* [33–35]. Onodera et al. showed that among UC patients, the C/D ratio of Tac was significantly lower in CYP3A5 expressers than in CYP3A5 non-expressers at 7–10 days after reaching a high trough level [36]. Hirai et al. also reported that CYP3A5-expressing patients with UC required a longer time to reach the effective blood concentration than CYP3A5 non-expressers [37]. Consistent with these previous findings, in the present study, CYP3A5-expressers required a significantly longer duration and significantly higher Tac dose and C/D ratio than CYP3A5 non-expressers.

In addition, the effects of *CYP3A4* polymorphisms on the blood concentration and clinical efficacy of Tac are controversial, especially in patients with UC. In the present study, we revealed the frequency of *CYP3A4*1G* polymorphisms, which could affect the enzyme activity of CYP3A4 [38] and regulate the metabolism of several drugs in Japanese patients with UC, and their correlation with the pharmacokinetics and clinical efficacy of Tac. The allele frequency of *CYP3A4*1G* was 19%, which is consistent with that reported previously, that is, 18.8%–23% in the Chinese population [39, 40] and 24.9% in the Japanese population [22]. Although the frequency of *CYP3A4*1G* polymorphisms is relatively high in East Asia, there are only a few reports on the relationship between *CYP3A4*1G* polymorphisms and Tac pharmacokinetics. Uesugi et al. reported that the Tac C/D ratio of patients with CYP3A4*1/*1G transplanted liver was significantly lower in the first 1 week after surgery than in patients with CYP3A4*1/*1 [41]. Li et al. reported that the C/D ratio of Tac in patients with *CYP3A4*1/*1* was significantly higher on day 7 after renal transplantation than that in patients with the *CYP3A4*1G* allele [24]. In the present study, patients with the *CYP3A4*1G* allele required a significantly longer duration than those with *CYP3A4*1/*1*. The C/D ratio was significantly lower in patients with the *CYP3A4*1G* allele than in those with *CYP3A4*1/*1*.

Regarding the effect of these gene polymorphisms on the therapeutic efficacy of Tac, the response rate at 1 week after Tac initiation in the present study was significantly higher in patients with UC with *CYP3A5*3/*3* than in those with the *CYP3A5*1* allele, consistent with the findings of a previous study [37] although some studies have not shown significant results in terms treatment efficacy [32, 36, 42]. Furthermore, in the present study, patients with *CYP3A4*1/*1* showed a higher response rate than those with *CYP3A4*1G*. Therefore, we concluded that the *CYP3A5*3* polymorphisms are independent predictors of early therapeutic effects.

However, in accordance with the findings of previous studies [18, 22], our study showed that *CYP3A5*3* and *CYP3A4*1G* polymorphisms had a strong LD relationship. Because of this LD, it is difficult to accurately evaluate the effect of *CYP3A4*1G* polymorphism on the pharmacokinetics and clinical efficiency of Tac. These two polymorphisms are not perfect, but are strongly related to each other. In fact, the *CYP3A4*1G* polymorphism has been reported to

contribute to the difference of individual in Tac pharmacokinetics among CYP3A5 expressers [18] and could help regulate the amount of Tac among CYP3A5 non-expressers [24]. In the present study, all CYP3A5 expressers carried the *CYP3A4*1G* allele. Moreover, among 15 patients with the *CYP3A4*1G* allele, CYP3A5 expressers showed a significantly lower C/D ratio than CYP3A5 non-expressers, suggesting that *CYP3A5* polymorphisms could have a stronger effect on Tac pharmacokinetics. However, because of the small sample size of the present study, further large-scale studies are needed to better understand this metabolic pathway.

Regarding safety, although several studies have shown that the *CYP3A5*3* polymorphism is related to nephrotoxicity, there is not enough consensus [42–46]. Asada et al. reported that the incidence of overall adverse events and nephrotoxicity was significantly higher in CYP3A5 expressers than in CYP3A5 non-expressers [42]. In the present study, there was no significant relationship between adverse events, including nephrotoxicity, and *CYP3A5*3* polymorphism, whereas patients with the *NR1I2–25385C/C* genotype presented significantly more overall adverse events than those with the *NR1I2–25385T* genotype. However, the mechanism cannot be explained by our findings as this polymorphism and adverse events were not related to the blood Tac trough levels. *NR1I2* 63396TT polymorphisms have been reported to be a risk factor for peripheral neuropathy in patients co-infected with human immunodeficiency virus and *Mycobacterium tuberculosis* [47]. Li et al. revealed that severe liver injury could be caused by certain drugs via PXR-mediated alteration of the heme biosynthesis pathway in PXR (encoded by *NR1I2*)-humanized mice [48]. Therefore, further functional studies on the effect of *NR1I2* on Tac treatment might provide useful information.

The present study had some limitations. First, this was a retrospective single-center study that comprised a small number of patients. The statistical analysis of the relationship between pharmacokinetics and *CYP3A4*1/*1* polymorphism along with *CYP3A4*1G* polymorphism could not be performed owing to the small number of patients. Second, measurement immunoassays of blood Tac concentration changed over the study period, which might affect concentration data. Third, the power to detect adverse events was insufficient because we only examined medical records. Fourth, except *CYP*, *NR1I2*, and *ABCC*, we did not investigate other enzymes involved in Tac metabolism, which might lead to missing potential confounding SNPs.

In conclusion, although we found that *CYP3A4*1G* and *CYP3A5*3* polymorphisms were related to Tac pharmacokinetics, *CYP3A5* polymorphism could have a stronger effect than *CYP3A4*, suggesting that these polymorphisms can be used to predict the short-term therapeutic effects of Tac. The *NR1I2–25385C/C* genotype was related to overall adverse events, whereas *CYP3A4*1G* and *CYP3A5*3* were not, suggesting that *NR1I2* polymorphism might be a potential predictor of the adverse events caused by Tac therapy. The evaluation of these polymorphisms could provide useful information on the status of Tac treatment in patients with UC. However, due to the small number of cases in this study, we hope that further external verification of clinical usefulness will be performed.

## Supporting information

**S1 Table. Combination of CYP3A4 and CYP3A5 gene polymorphisms.**
(PDF)

**S2 Table. Adverse event summary.**
(PDF)

## Acknowledgments

All authors agree with this submission. SH and KW made substantial contributions to the concept of this study. MF, SH, YN, and SI made substantial contributions to data and sample acquisition. MF and SH made substantial contributions to the data analysis. MF and SH made substantial contributions to manuscript writing. Yuji Nadatani, SH, KO, FN, Yasuaki Nagami, KT, NK, TW, KW, and YF made substantial contributions to reviewing and editing this manuscript.

## Author Contributions

**Conceptualization:** Shuhei Hosomi, Kenji Watanabe.

**Data curation:** Maizumi Furuse, Shuhei Hosomi, Yu Nishida, Shigehiro Itani.

**Formal analysis:** Maizumi Furuse, Shuhei Hosomi.

**Writing – original draft:** Maizumi Furuse.

**Writing – review & editing:** Shuhei Hosomi, Yu Nishida, Shigehiro Itani, Yuji Nadatani, Shusei Fukunaga, Koji Otani, Fumio Tanaka, Yasuaki Nagami, Koichi Taira, Noriko Kamata, Toshio Watanabe, Kenji Watanabe, Yasuhiro Fujiwara.

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
