## [Decision Letter · Decision Letter 0]

17 Feb 2021

PONE-D-20-34236

The impact of *cytochrome P450 3A* genetic polymorphisms on tacrolimus pharmacokinetics in ulcerative colitis patients

PLOS ONE

Dear Dr. Shuhei Hosomi,

Thank you for submitting your manuscript to PLOS ONE. After careful consideration, we feel that it has merit but does not fully meet PLOS ONE’s publication criteria as it currently stands. Therefore, we invite you to submit a revised version of the manuscript that addresses the points raised during the review process.

Overall the manuscript covers an area of interest in the field of tacrolimus pharmacogenetics and even if exploratory the results reported could be interesting for the journal's readers.

Since the number of patients included in the paper is very low, I would reccomend to highlight the exploratory nature of the study end the necessity to perform further external validation of the results before translating them into clinical prescription

We look forward to receiving your revised manuscript.

Kind regards,

Erika Cecchin

Academic Editor

PLOS ONE

Journal Requirements:

2. In your ethics statement in the manuscript and in the online submission form, please provide additional information about the patient records used in your retrospective study. Specifically, please ensure that you have discussed whether all data were fully anonymized before you accessed them and/or whether the IRB or ethics committee waived the requirement for informed consent. If patients provided informed written consent to have data from their medical records used in research, please include this information.

- https://www.futuremedicine.com/doi/10.2217/pgs.11.33

- https://www.jstage.jst.go.jp/article/bpb/36/11/36_b13-00509/_html

- https://academic.oup.com/ibdjournal/article/15/1/134/4647657

In your revision ensure you cite all your sources (including your own works), and quote or rephrase any duplicated text outside the methods section. Further consideration is dependent on these concerns being addressed.

Reviewers' comments:

Reviewer's Responses to Questions

**Comments to the Author**

1. Is the manuscript technically sound, and do the data support the conclusions?

Reviewer #1: Yes

Reviewer #2: Yes

2. Has the statistical analysis been performed appropriately and rigorously? 

Reviewer #1: Yes

Reviewer #2: Yes

3. Have the authors made all data underlying the findings in their manuscript fully available?

Reviewer #1: Yes

Reviewer #2: Yes

4. Is the manuscript presented in an intelligible fashion and written in standard English?

Reviewer #1: Yes

Reviewer #2: Yes

5. Review Comments to the Author

Reviewer #1: The authors investigated the influence of cytochrome P450 3A4/5, 2C19, ABCC2, and NR1I2

genetic polymorphisms on tacrolimus pharmacokinetics, ulcerative colitis severity before tacrolimus therapy, and adverse events during treatment in Japanese ulcerative colitis patients. Genetic variation in cytochrome P450 3A4/5 as well as other genes are known to influence tacrolimus pharmacokinetics, but there remains variability in tacrolimus pharmacokinetics that has not been explained. It is imperative that tacrolimus pharmacokinetics be predictable in order to treat ulcerative colitis effectively, and the authors present a candidate polymorphism study to identify genetic variants that can be used to predict tacrolimus pharmacokinetics and response.

1. On page 2 (line 26, 32, and 34), page 13 (line 199), page 14 (line 208), and page 16 (line 262, 263, and 278), the authors specify the genomic position of the nuclear receptor subfamily 1 group I member 2 polymorphism as -1570 when referencing the polymorphism, (NR1I2)-1570C>T. There is a discrepancy between the genomic location in the above sentence (-1570) and tables 2, 3, and 6 where the position is reported as -25385. The polymorphism is also referred to as (NR1I2)-25385C>T on page 4 (line 65 and 66).

2. On page 2 (line 26), page 6 (line 100), the authors specify the genomic position of the ATP-binding cassette subfamily C member 2 polymorphism as position -26. This does not agree with the genomic position reported in tables 2, 3, and 6 as -24.

3. There is a small phrasing error on line page 3 (line 49). The sentence “An SNP at position 6986A…..” would benefit from being changed to “A SNP at position 6986A…..”

4. On page 5 (line 89), please include a short description or definition of the partial Mayo score to help the reader. This would make things easier for the reader instead of having the reader refer to reference [29] for a description.

5. On page 6 (line 107), it is not clear what P-value cutoff was chosen for deviation from Hardy-Weinberg equilibrium. It can be assumed that the cutoff chosen was p<0.05 based on the sentence on page 7 (line 115-116), but it is not entirely clear to the reader.

6. On page 6 (line 110-111), please clarify how the statistical test of Chi-squared or Fisher’s exact test was decided on (e.g., “Fisher’s exact test for small sample size” would suffice). It is not clear to the reader how the authors decided to use the chi-squared or Fisher’s exact test.

7. In table 3, it would be easier for the reader if the grey and white color scheme in the table were alternated by allele as in table 2 instead of by gene. This would make it easier for the reader to identify the two allele groups that were compared for each statistical test shown in the table.

8. In table 3, 4, 5, 6, and S1 it would be easier for the reader to identify significant associations if asterisks indicating level(s) of significance were included as a superscript on the P values (e.g., * = P<0.05, ** = P<0.01)

Reviewer #2: The study was done on a low number of patients (45) but it could provide useful information in the treatment of UC with tacrolimus. It is not clear to me whether the large SD are due to the method used to do the pharmacokinetics measurements. The samples were collected between 2008 and 2018 and the technique(s) used to measure the Tac levels may have changed over this time period.

6. PLOS authors have the option to publish the peer review history of their article (what does this mean?). If published, this will include your full peer review and any attached files.

Reviewer #1: No

Reviewer #2: No

---

## [Author Response · Author response to Decision Letter 0]

8 Mar 2021

Responses to Academic Editor:

Thank you very much for your excellent suggestion. We have added the following text to the last paragraph of the Discussion section: “due to the small number of cases in this study, we hope that further external verification of clinical usefulness will be performed”

Responses to Journal Requirements:

1. We have ensured that our manuscript meets PLOS ONE's style requirements, including those for file naming.

2. We have added “Ethical considerations” to the method section.

“Ethical considerations

This study was approved by the Ethics Committee of Osaka City University Graduate School of Medicine (approval number: 3293). Written informed consent was obtained from all patients at the start of this study. All data were fully anonymized before we accessed them.”

3. We have rephrased these duplicate points in the Discussion section. 

“The C/D ratio of Tac in UC patients with the CYP3A4*1G allele was statistically lower than in those with the CYP3A4*1/*1 allele” on page 2.

“Approximately 15% of UC patients have been reported to develop an acute severe colitis, which sometimes requires urgent / emergency surgery owing to major complications such as perforation, toxic megacolon, and massive hemorrhage[30]. “ on page 12.

“Uesugi et al. reported that the Tac C/D ratio of patients with CYP3A4*1/*1G transplanted liver was significantly lower in the first 1 week after surgery than in patients with CYP3A4*1/*1[41]” on page 14.

“These two polymorphisms are not perfect, but are strongly related to each other. In fact, the CYP3A4*1G polymorphism has been reported to contribute to the difference of individual in Tac pharmacokinetics among CYP3A5 expressers[18].” on page 14.

“In the present study, all CYP3A5 expressers carried the CYP3A4*1G allele.” on page 14.

 

Responses to reviewer #1:

Thank you for your constructive suggestions. We performed some corrections based on your comments. All changes in the manuscript in response to the critiques are indicated with yellow highlights.

1. On page 2 (line 26, 32, and 34), page 13 (line 199), page 14 (line 208), and page 16 (line 262, 263, and 278), the authors specify the genomic position of the nuclear receptor subfamily 1 group I member 2 polymorphism as -1570 when referencing the polymorphism, (NR1I2)-1570C>T. There is a discrepancy between the genomic location in the above sentence (-1570) and tables 2, 3, and 6 where the position is reported as -25385. The polymorphism is also referred to as (NR1I2)-25385C>T on page 4 (line 65 and 66).

We appreciate your suggestion. As for the genomic position of the nuclear receptor subfamily 1 group I member 2 (NR1I2) polymorphism of dbSNP 3814055, we have confirmed and unified it to NR1I2-25385. Therefore, we modified from NR1I2–1570 to NR1I2–25385 on page 2, 12, 15, and 16.

2. On page 2 (line 26), page 6 (line 100), the authors specify the genomic position of the ATP-binding cassette subfamily C member 2 polymorphism as position -26. This does not agree with the genomic position reported in tables 2, 3, and 6 as -24.

We thank the reviewer for the suggestion. As for the genomic position of the ATP-binding cassette subfamily C member 2 (ABCC2) polymorphism of dbSNP 717620, we have confirmed and unified it to ABCC2–24. Therefore, we modified from ABCC2–26 to ABCC2–24 on page 2 and 5.

3. There is a small phrasing error on line page 3 (line 49). The sentence “An SNP at position 6986A…..” would benefit from being changed to “A SNP at position 6986A…..”

Thank you for noticing the phrasing error. We changed the sentence “An SNP at position 6986A…..” to “A SNP at position 6986A…..”.

4. On page 5 (line 89), please include a short description or definition of the partial Mayo score to help the reader. This would make things easier for the reader instead of having the reader refer to reference [29] for a description.

We fully agree with your comment. We have added the text “the partial Mayo score (sum of 3 subscores of the Mayo score without the endoscopic findings)” on page 5.

5. On page 6 (line 107), it is not clear what P-value cutoff was chosen for deviation from Hardy-Weinberg equilibrium. It can be assumed that the cutoff chosen was p<0.05 based on the sentence on page 7 (line 115-116), but it is not entirely clear to the reader.

We appreciate your constructive comment. To help the reader understand the selected P-value cutoff for deviations from Hardy-Weinberg equilibrium easily, we have added the sentence of “p > 0.05 (chi-squared test) was considered to indicate equilibrium” on page 6.

6. On page 6 (line 110-111), please clarify how the statistical test of Chi-squared or Fisher’s exact test was decided on (e.g., “Fisher’s exact test for small sample size” would suffice). It is not clear to the reader how the authors decided to use the chi-squared or Fisher’s exact test.

Thank you for your valuable suggestion. In the Statistical analysis sub-section in the Materials and methods, we added the sentence "Fisher's exact test was applied to small samples." to clarify to the reader how we decided to use the chi-square test or Fisher's exact test.

7. In table 3, it would be easier for the reader if the grey and white color scheme in the table were alternated by allele as in table 2 instead of by gene. This would make it easier for the reader to identify the two allele groups that were compared for each statistical test shown in the table.

As the reviewer suggested, the gray and white color schemes in the table have been modified to alternate by allele in Table 3. We agree that this change would be better. 

8. In table 3, 4, 5, 6, and S1 it would be easier for the reader to identify significant associations if asterisks indicating level(s) of significance were included as a superscript on the P values (e.g., * = P<0.05, ** = P<0.01)

As the reviewer suggested, we added asterisks indicating level(s) of significance as a superscript on the P values (# = P < 0.05, ## = P < 0.01, ### = P < 0.001) in Tables 3, 4, 5, 6, and S1 to help the reader to clearly identify important relevance. 

Responses to reviewer #2:

The study was done on a low number of patients (45) but it could provide useful information in the treatment of UC with tacrolimus. It is not clear to me whether the large SD are due to the method used to do the pharmacokinetics measurements. The samples were collected between 2008 and 2018 and the technique(s) used to measure the Tac levels may have changed over this time period.

Thank you for providing these insights. As the reviewer pointed out, measurement assays of blood Tac concentration changed over the study period. Therefore, we have added the information in the Treatment protocol and evaluation of treatment efficacy sub-section of the Materials and methods. 

Blood Tac concentration was measured by either an affinity column-mediated immunoassay (from January 2009 to January 2013), a chemiluminescent immunoassay (from February 2013 to June 2014), or an electro‐chemiluminescence immunoassay (July 2014 to January 2018) in the in-hospital laboratory.

Each immunoassay has been reported to be well correlated [Shigematsu T, et al. Ther Drug Monit. 2020 Jun;42(3):400-406], especially a chemiluminescent immunoassay and an electro-chemiluminescence immunoassay with high sensitivity [Miura M, et al. Biol Pharm Bull. 2016 Aug 1;39(8):1331-7.], and no particular trend was observed in the scatter of the concentration/dose (C/D) ratio in the present study. 

However, this could be a potential limitation affecting the C/D ratio, therefore, we have added this in the limitation. 

“Second, measurement immunoassays of blood Tac concentration changed over the study period, which might affect concentration data.” on the limitation in the Discussion.

Despite the limitation, we believe that our results could provide useful information on the status of Tac treatment in patients with UC.

---

## [Decision Letter · Decision Letter 1]

12 Apr 2021

The impact of *cytochrome P450 3A* genetic polymorphisms on tacrolimus pharmacokinetics in ulcerative colitis patients

PONE-D-20-34236R1

Dear Dr. Shuhei Hosomi,

We’re pleased to inform you that your manuscript has been judged scientifically suitable for publication and will be formally accepted for publication once it meets all outstanding technical requirements.

Kind regards,

Erika Cecchin

Academic Editor

PLOS ONE

Additional Editor Comments (optional):

Reviewers' comments:

Reviewer's Responses to Questions

**Comments to the Author**

1. If the authors have adequately addressed your comments raised in a previous round of review and you feel that this manuscript is now acceptable for publication, you may indicate that here to bypass the “Comments to the Author” section, enter your conflict of interest statement in the “Confidential to Editor” section, and submit your "Accept" recommendation.

Reviewer #1: All comments have been addressed

Reviewer #3: (No Response)

2. Is the manuscript technically sound, and do the data support the conclusions?

Reviewer #1: Yes

Reviewer #3: Yes

3. Has the statistical analysis been performed appropriately and rigorously? 

Reviewer #1: Yes

Reviewer #3: Yes

4. Have the authors made all data underlying the findings in their manuscript fully available?

Reviewer #1: Yes

Reviewer #3: Yes

5. Is the manuscript presented in an intelligible fashion and written in standard English?

Reviewer #1: Yes

Reviewer #3: Yes

6. Review Comments to the Author

Reviewer #1: Thanks to the authors in addressing my concerns on this interesting manuscript, I have no additional comments.

Reviewer #3: In this article the authors identify some polymorphisms that could be useful to optimize the administration of tacrolimus in patients with refractory ulcerative colitis. The main limitation of the study is the very small sample size of the study population that negatively impact the statistical power of the work. However, the authors have discussed this point in the limitation section. The topic could be of interest and may provide new insight in the definition of predictive genetic markers that could be used in clinical practice to individualize and optimize the tacrolimus treatment. Globally the work could be accepted.

7. PLOS authors have the option to publish the peer review history of their article (what does this mean?). If published, this will include your full peer review and any attached files.

Reviewer #1: No

Reviewer #3: No

---

## [Editor Report · Acceptance letter]

14 Apr 2021

PONE-D-20-34236R1 

The impact of *cytochrome P450 3A* genetic polymorphisms on tacrolimus pharmacokinetics in ulcerative colitis patients 

Dear Dr. Hosomi:

I'm pleased to inform you that your manuscript has been deemed suitable for publication in PLOS ONE. Congratulations! Your manuscript is now with our production department. 

Kind regards, 

on behalf of

Dr. Erika Cecchin 

Academic Editor

PLOS ONE